# Factors Influencing Parental Awareness about Newborn Screening

**DOI:** 10.3390/ijns5030035

**Published:** 2019-09-18

**Authors:** Věra Franková, Alena Dohnalová, Karolína Pešková, Renata Hermánková, Riona O’Driscoll, Pavel Ješina, Viktor Kožich

**Affiliations:** 1Department of Pediatrics and Adolescent Medicine, Charles University First Faculty of Medicine and General University Hospital, Ke Karlovu 2, 128 04 Prague 2, Czech Republic; karolina.peskova@vfn.cz (K.P.); renata.hermankova@vfn.cz (R.H.); riona.od@gmail.com (R.O.); pavel.jesina@vfn.cz (P.J.); 2Institute for Medical Humanities, First Faculty of Medicine, Charles University, Karlovo náměstí 40, 128 08 Prague 2, Czech Republic; 3Institute of Physiology, First Faculty of Medicine, Charles University, Albertov 5, 128 00 Prague 2, Czech Republic; alena.dohnalova@lf1.cuni.cz

**Keywords:** newborn screening, education, consent, survey, parental awareness

## Abstract

Appropriate and timely education about newborn screening (NBS) helps to foster benefits such as prompt follow up, to promote parents’ autonomy via informed consent and minimize the harms such as reducing the impact of NBS false-positive results. The aim of this study was to ascertain how mothers are informed about NBS in the Czech Republic and to identify the variables associated with awareness about NBS. The questionnaires evaluating awareness and its determinants were mailed to a random sample of 3000 mothers 3 months post-delivery. The overall response rate was 42%. We analysed 1100 questionnaires and observed that better awareness about NBS was significantly associated with age, parity, number of information sources, child health status, size of maternity hospital and an obstetrician as the source of prenatally obtained information. Although the majority of mothers (77%) in our study recalled being informed by a physician or nurse in the neonatal ward, results have revealed that over 40% of participants did not have sufficient awareness about the principal aspects of NBS. Several measures including seminars for healthcare providers and the development and distribution of new educational materials were adopted to improve parental education about NBS in the Czech Republic.

## 1. Introduction

Newborn screening (NBS) is a preventive public health programme carried out globally. In the early 1960s NBS was introduced in the US and since then has expanded to many countries around the world. Although there are large variations in the panels of screened disorders and the set-up of individual programmes [1] the principles and goals of these programmes are the same. The aim is to identify infants with increased risk of certain rare disorders and to provide them with early treatment and other interventions which can prevent serious damage to their health. These benefits can only be achieved with prompt follow up to confirm or refute a presumptive diagnosis after an initial positive result. In order for this to happen parents need to be informed about the nature of NBS and its purpose. Giving parents adequate information about NBS prior to testing may also help to reduce parental distress in the event of a positive or false positive screening result [2,3,4,5]. Therefore parental education about NBS and consent for screening should be an integral part of the NBS programme. The approaches to parental consent differ considerably between countries, ranging from a mandatory NBS programme to an opt-out approach and explicit consent [6,7,8].

The need to improve parental education has been stressed particularly after the introduction of tandem mass spectrometry (MS/MS) into NBS programmes in the 1990s. This has led to the expansion of screening panels from only a few to dozens of disorders. Since then it has been documented that parental education about NBS is not effective; parents are not receiving the information or are unware about screening and its principal aspects [9,10,11,12]. The last trimester of pregnancy has been suggested as the optimal period for information provision [1], but a survey carried out among EU member states revealed that 45% of them inform parents only after birth at the time of blood sampling [6]. This may reflect the ongoing perception of NBS being a routine test performed as part of standard postnatal care rather than specifically a screening test that parents should be adequately informed about before consenting to the procedure. 

The NBS programme in the Czech Republic is not regulated by any specific legally binding provisions, there are only national guidelines issued by the Czech Ministry of Health [13]. This is similar to many European countries were the actual screening process adheres to certain guidelines but is not specifically regulated by them [6]. The current Czech programme includes a panel of 18 disorders. Regular nationwide NBS screening for phenylketonuria started in the former Czechoslovakia in 1975. Since then several different disorders have gradually been added to the panel which now includes 2 endocrine disorders, 15 metabolic disorders, and cystic fibrosis. A web site for the public and health professionals was established in 2009 [14]. Czech national guidelines do not explicitly specify the timing for provision of NBS information [13]. According to the Act on Healthcare Services [15], NBS can be performed like other medical procedures with only parental verbal or written consent after appropriate information provision. The healthcare provider (i.e., maternity hospital) can choose which form of consent is used. Written documentation is required only from those parents who opt out of NBS. The long-term practice is to inform parents about NBS during the postnatal period usually shortly before sampling. As midwifes are not involved in education about NBS this information is usually provided by physicians or nurses in the neonatal ward. With regard to reporting of results, the policy of the Czech NBS programme is to only issue reports in the event of a positive result or if a repeat card is required. Reports are not issued on screen negative results therefore parents should be made aware they will not be informed personally of results.

The aim of this survey study was to map how mothers are informed about NBS in the Czech Republic. We also wanted to identify the variables associated with the awareness about NBS and explore the most common sources of information. It was anticipated that the findings from this study could be used for future improvements to the provision of NBS information to parents and increase their awareness of the NBS process and its importance. 

## 2. Materials and methods

### 2.1. Sample

A random sample of 3000 mothers whose newborns’ blood samples for NBS were analyzed in the Institute of Inherited Metabolic Disorders (General University Hospital in Prague and the Charles University in Prague, First Faculty of Medicine, Czech Republic) were invited to participate in the study. The questionnaires and the invitation letter explaining the design and aims of the study were mailed to mothers 3 months post-delivery between January and February 2014. Mothers were asked to return completed questionnaires by mail using the stamped envelope included in the mailed package. Return of the questionnaires was taken as an indication of consent to participate in this study. Questionnaires did not have any identification code and their evaluation was completely anonymous. This study received approval from the Ethics Committee of the General University Hospital in Prague (approval number 1937/13 S-IV, approval date 15 August 2013).

### 2.2. Questionnaires

We developed two questionnaires for this study; one for evaluating awareness about NBS and identifying sources of information and the second one to determine demographic and other factors which might influence the level of awareness about NBS. These two questionnaires were developed after multiple rounds of discussion among team members with expertise in newborn screening, genetic counselling, paediatrics, medical ethics, psychology and survey research methods. After a review of the literature including existing recommendations on parental education in the Czech language [13,14], members of the study team proposed questions and identified possible variables that might influence the awareness about NBS. These were then reviewed and approved by the whole study team. We asked a small group of mothers of paediatric patients of the Institute of Inherited Metabolic Disorders to evaluate the clarity of the questionnaires. This led to minor revisions consisting mostly of rephrasing the surveyed items.

The final version of the questionnaire for the evaluation of awareness about NBS consisted of questions about: (i) NBS itself (knowledge that NBS is performed on all newborns, the purpose of NBS, sample timing, screened diseases and the possibility of false positive results); (ii) identification of sources of information with the option to select multiple sources (obstetrician, physician at neonatal clinic/maternal ward, paediatrician, other sources and not informed); and (iii) active use of the internet. For the evaluation of the awareness about NBS we assigned points to each question according to their relative importance: knowledge that NBS is performed on all newborns (the fundamental principle of screening–4 points), the purpose of NBS (important knowledge–2 points), the possibility of a false positive result (important knowledge–2 points), sample timing (less important technical fact–1 point), screened diseases (less important technical fact–1 point. A false positive result was defined as what occurs when the first result of the newborn screening test suggests the infant may have a certain condition which is subsequently excluded after further examination of the infant. The score for the NBS awareness ranged from minimum of 0 to a maximum of 10 points.

To find out the determinants possibly influencing the awareness about NBS, the final version of the second questionnaire included 8 items which consisted of: (i) basic demographic data of participants (age, parity, self-reported health status, education and the population of place of residence); (ii) complications during the pregnancy; (iii) size of maternity hospital; and (iv) child health status.

Participants were asked to add free text comments on their experience with NBS education such as how the information about NBS was provided and the NBS programme in general. Free text comments were analyzed for common topics by two independent observers. 

### 2.3. Data Analysis

Collected data were analysed using SPSS 13.0 statistics software (SPSS Inc., Chicago, USA). Demographic data of the study sample were compared against the data available in the Report on Mother and Newborn 2013 issued by the Institute of Health Information and Statistics of the Czech Republic [16].

After analysis using descriptive statistics the total scores from the questionnaire evaluating awareness about NBS were counted. Determinants of the total awareness score were tested by the non-parametric Mann-Whitney and Kruskal-Wallis tests for binary and multiple variables, respectively. The Mann-Whitney test was also used for testing whether there was a difference between participants informed from a single source and participants informed from multiple sources and whether participants informed by an obstetrician scored higher in the questionnaire evaluating awareness about NBS than other participants. Chi-squared test (χ^2^–test) was used to assess binary categorical variables. Statistical significance was set at *p* < 0.05.

## 3. Results

### 3.1. Response Rate and Sample Characteristics

The overall response rate was 42% (1162/2793). From the initial sample of 3000 mailed questionnaires 207 were not delivered due to an incorrect address. We received questionnaires from 1162 respondents; 62 were incomplete and therefore excluded from the final analysis, which was performed on questionnaires from 1100 participants. 

The majority of participants were over 30 years of age (69%) (Table 1), the age range was from 18 to 43 years, with a mean age of 31.7. Nearly half of the respondents had a university degree (48%) and more than one child (49%) at the time of the study (Table 1). Most of participants did not have health problems (91%) at the time of the study, did not have complications during the pregnancy (79%) and did not have a child with a long term serious illness (98%). Table 1 also shows a comparison of sample characteristics to data available from the Report on Mother and Newborn 2013 [16] in respect to age, education, parity, complications during pregnancy, child health status and size of maternity hospital. Considerable differences between the study sample and the general maternal population were found in education (48% vs. 29% of mothers with a university degree) and age (69% vs. 55% of mothers ≥ 30 years old) (Table 1). The study sample was on average almost two years older than the general maternal population (mean age 31.7 vs 29.9) [16].

### 3.2. Awareness about NBS

Participants were asked to record whether at the time of sampling they were familiar with the information about NBS contained in the questionnaire. More than two thirds of participants indicated that they knew: (a) NBS is performed on all newborns in the Czech Republic (70%); (b) the purpose of NBS (68%) and (c) were familiar with the appropriate time of sampling (67%). Much fewer participants indicated that they knew the screened conditions in NBS (37%) and about the possibility of false-positive results (30%) (Figure 1).

The overall score from the NBS awareness questionnaire was evaluated for each participant. The score could range from a minimum of 0 to a maximum of 10 points. Figure 2 shows the distribution of the total scores among participants. Surprisingly 18% of participants (*n* = 197) reported not knowing anything about NBS (score 0). The largest proportion of participants (20%, *n* = 223) reached a score of 7 points. This score might be considered as sufficient awareness about NBS; to achieve 7 points a participant had to be familiar with the fact that NBS is performed on all newborns (4 points) and answer positively to at least two more questions (including at least one of the more relevant questions for 2 points) (Figure 1). A score of 7 or more points was reached by more than half of participants (58%) and about 16% (*n* = 180) were aware of all aspects of NBS surveyed in the questionnaire (score 10 points). These results show bimodal distribution of participants’ awareness about NBS; they were either both well informed and retained the information or their knowledge about NBS was rather poor. 

### 3.3. Source of Information about NBS 

Participants were asked to indicate all sources from which they received information about NBS. A surprisingly high number of participants (11%) reported that nobody informed them about NBS. The majority was provided with information about NBS by a physician (68%) or a nurse (9%) in the neonatal ward (Figure 3.). Very few were informed by an obstetrician (5%) during the prenatal period or by a paediatrician (3%). A number of participants (17%) indicated in written comments that they had learnt about NBS from other sources including educational materials from maternity hospital (3%), prenatal courses (3%), magazines (3%), books (2%), friends (2%), family members (1%), during a previous pregnancy (2%) or during their professional training (3%). Information about NBS was actively sought on the internet by 22% of participants (Figure 3.). 

Figure 4 indicates how many sources participants used to obtain NBS information. The majority of participants (65%) received information from only one source; 27% used 2 or more informational sources whilst 8% of participants did not receive information from any source and also did not seek information from any other sources. Of the participants informed by an obstetrician (*n* = 55), 78% of these (*n* = 43) were also informed from additional sources.

### 3.4. Factors Associated with the Awareness about NBS 

Statistical analysis demonstrated that significantly higher scores in the questionnaire evaluating NBS awareness were reached by participants who were≥30 years (*p* < 0.001), had more than one child (*p* < 0.001) and obtained information from multiple sources (*p* < 0.001) (Table 2). Sixty percent of mothers who were≥30 years were also multiparous. Participants informed about NBS during the prenatal period by an obstetrician were also more likely to have a higher score (*p* < 0.001) (Table 2). The participants’ level of education and the population size of the place of residence did not show statistically significant difference in scores (Table 2). 

We assumed that education about NBS might vary in relation to the size of the maternity hospital, which we defined by number of births per year (Table 1). In statistical analysis, significantly higher scores (*p* < 0.05) were reached by participants with delivery in the middle sized maternity hospital (1000–3000 births/year) (Table 2). Participants who reported that nobody informed them about NBS (*n* = 119) were significantly more often (*p* < 0,001) from the large maternity hospitals (>3000 births/year) than from the middle sized and the small maternity hospitals (<1000 births/year) (tested by χ^2^-test, results not shown). Those who reported not being informed of NBS from any source had the lowest scores in the questionnaire evaluating awareness. Participants who opted for a homebirth (*n* = 3) reached a high score in the questionnaire evaluating NBS awareness (mean = 8.00). However due to the very low number in this specific group, it was not included in the statistical analysis of the place of delivery. 

We also included three health-related variables (complications during the pregnancy, current self-reported health status of participants, and the child health status) since we assumed that these variables might influence the information retrieved on the internet and the search for medical information generally. Statistically significant lower scores were obtained by participants with a child with a long-term serious illness (most frequently congenital malformations) in comparison to participants with a healthy child (Table 2). The difference is most likely caused by concerns about child health and possibly the large number of different medical examinations performed shortly after childbirth. 

### 3.5. Analysis of the Free Text Comments about NBS Education and the NBS Programme

A substantial number of participants (397) used the possibility to add free text comments concerning NBS education and the NBS programme generally. Most of the participants stated in their comments that they had a limited awareness about NBS. With regards to education about NBS participants often mentioned poor explanation of the nature of the NBS process and no specific information about the possibility of false-positive results. During their stay in the maternity hospital they were usually informed only about the heel prick and instructed to ask a paediatrician about the result. Some participants were not sure if they were informed at all but they often stated that they “maybe just do not remember due to other concerns”. Fewer participants were satisfied with the information provided and “did not want to know more details about NBS unless the child would suffer from such a disease”. Participants stated that education about NBS should take place before delivery optimally during the prenatal period and specifically suggested that education about NBS should be provided directly by a conversation with a health care professional together with an information leaflet which parents can refer to at a later stage. 

The most common topic in relation to NBS generally was the communication of NBS results. Participants felt that they were not informed about NBS results and expressed a desire to receive a hard copy. This comment suggests that the participants were not adequately informed about the set-up of the NBS programme where negative results are not reported either to clinicians or parents. They are only informed in the event of a positive result or a recall card. Some participants expressed their satisfaction with NBS as the “opportunity to examine the child for the diseases included in NBS”. Participants also valued the current study as “the possibility for future improvement”.

## 4. Discussion

This study is the first attempt to ascertain how mothers are informed about NBS in the Czech Republic and to identify the variables associated with NBS awareness, which may serve as the starting point for future improvements. Results have shown that Czech mothers in the study sample have only limited NBS awareness and three months post-delivery around one fifth of mothers do not recall any information about NBS. Limited awareness and knowledge about NBS has been demonstrated by previous international studies that have focused on education about NBS [10,11,17,18,19,20,21,22,23], the informed consent process [7,24,25,26], and the impact of positive and/or false positive NBS results [2,4,5,9,27]. These studies have been performed in Canada [7,11], the US [2,3,4,5,18,20], the UK [22,23,25,26], Australia [10,17], the Netherlands [24] and Saudi Arabia [21]. Thus in comparison to the Czech Republic they studied populations with different socio-cultural backgrounds and also a different organisational structure of their NBS programmes. This includes how NBS information is provided before testing and the method of obtaining parental consent for NBS. Despite these differences the studies revealed similar results to our study specifically in relation to some mothers/parents being unable to recall any information regarding NBS [4,11,21,25] and feeling not well informed [3,10,11,20,24,25]. 

Our study results have shown that better awareness about NBS has been positively associated with older age, higher parity, number of information sources, good child health status, medium sized maternity hospitals and an obstetrician as the source of prenatal information (Table 2). Other variables (level of education, place of residence, complications during the pregnancy and self-reported health status after pregnancy) did not show significant differences in awareness about NBS. Unsurprisingly, multiparous mothers and mothers’≥30 years of age were better informed about NBS. This result can be explained by their own previous experiences with NBS or by the experience of their relatives and friends. Previous experience with NBS has been also shown as a significant predictor of perceived informed choice about NBS [26]. In a study by Araia et al. receiving information prenatally has been associated with increased knowledge of NBS [11]. Only 5% of our participants were provided with the information by an obstetrician during the prenatal period and this group of mothers were significantly better informed and retained the information (Table 2). The majority of mothers in this group (78%) used additional sources of information and were probably more informed about NBS prior to sampling. Obtaining the information from multiple sources has been associated with significantly better awareness about NBS in the whole study population. Results also revealed that mothers in large maternity hospitals (>3000 births/year) were less aware about the principal aspects of NBS (Table 2) and were significantly more often uninformed. This may be due to a more routine approach and less individualized care in larger medical facilities. Factors associated with a higher knowledge or awareness about NBS in other survey studies [11,17] were either not found in this study (higher level of education) or were not investigated as a possible determinant (mothers’ income, language spoken at home). 

In comparison to other studies lower number of mothers in our study group (approx. 11% vs. 20% [25], 30% [11] and 35% [20]) did not recall being provided with any information about NBS at any time. Although the majority of mothers (77%) in our study recalled being informed by a physician or a nurse in the neonatal ward results have revealed that over 40% of participants did not have sufficient awareness about the principal aspects of NBS (Figure 2). These findings are most often explained by post-partum provision of information [3,10,17] and a routine approach to NBS [2,11,24]. The post-partum period when mothers are psychologically overwhelmed, physically exhausted and receive large amounts of information about neonatal care is considered as suboptimal to learn and retain NBS information [3]. In the Czech Republic the long-term practice is to inform parents about NBS within 72 h after birth which is usually just shortly before sampling. Some of the mothers’ written comments also implied that the post-partum period is not an appropriate time for education about NBS since some mothers were not sure if they were informed or they “just do not remember”. Therefore it is hard to determine whether mothers who reported that nobody informed them (Figure 3.) really had not been provided with the information, had not recognized it or just simply forgot it. 

The parental experience of NBS as a standard procedure or “the heel prick” rather than as a process to identify potentially affected children was reported by several qualitative studies [2,24,25] and confirmed by free text comments from our participants. This routine approach dates to the time when NBS programmes included only a small number of disorders with clear health benefits and minimal harms [3]. With the expansion of the NBS panel of tested disorders the probability of the associated harms including false positive results are increasing [28]. False positive results may generate substantial short-term parental anxiety and may be the most commonly identified psychosocial harm to families from NBS [29]. Appropriate education about NBS including information about the probability of positive and false positive results might decrease the parental anxiety and emotional distress [2,3,4,5]. In our study sample less than one third of mothers were aware of the possibility of a false positive NBS result. The absence of specific information about false positive results was also mentioned in written comments. With increased complexity in the NBS testing process the substitution of a routine approach to NBS with adequate education and parental consent (or dissent) to the procedure has been recommended internationally. However the evidence demonstrates slow progression towards appropriate parental NBS education and consent. [7]. The issue of explicitly informed parental consent is of particular importance with the emergence of next-generation sequencing technologies and its possible application to NBS testing. A crucial point to consider will be informing parents of this complex technology and the issues associated with such testing methods. This may require more time and resources for parental education and consent [30]. Having an adequate parental NBS consent and education framework in place prior to any such changes will greatly assist in the transition to NGS if this arises.

The provision of written material about NBS to parents is not obligatory therefore parents can be informed just orally by a healthcare professional. Only 3% of mothers in our study population had been provided with printed educational material at a maternity hospital Variations among individual maternity hospitals in the provision of the information may influence the overall parental awareness and understanding of NBS. The differences among maternity hospitals have been also observed in written comments. Some mothers stated they were satisfied with the education about NBS and the amount of information provided, however a substantial number mentioned the poor explanation of the nature of the NBS process. 

The most common source of information about NBS reported by our participants was the health care professionals at maternity hospitals (Figure 3). This finding is consistent with international studies [11,22]. Very few mothers in our study were informed about NBS prenatally by an obstetrician or during antenatal classes, which are above-standard care and paid for by prospective mothers. The second most frequent source of information was the internet (Figure 3). The official website of the Czech NBS programme provides reliable and up-to-date information which is also cited on the majority of other websites for the lay public dedicated to pregnancy and birth. Our participants identified a range of other more or less reliable information sources from which they had learnt about NBS but none of these were used by more than 3% of mothers (Figure 3). 

We acknowledge several limitations to this study. Although the response rate of 42 % is close to other survey studies (32–47%) regarding knowledge and awareness about NBS [10,11,26], nonresponse bias is likely to play some role in our findings. In comparison to data from the Report on Mother and Newborn 2013 [16], mothers in our study group were substantially older and had higher education (Table 1). Therefore we can hypothesize that mothers who choose to participate had a higher awareness of NBS than younger and less educated nonresponders. Our study group is also a retrospective sample therefore the time frame (three months post-delivery) may introduce an element of recall bias. The questionnaire evaluating awareness did not contain control questions therefore only self-reported knowledge was assessed. Another limitation is that regression analysis could not be used as the scores from the awareness questionnaire were non-linear thus preventing assessment of the effect of demographic characteristics and their interaction. Despite these limitations, our results provide insight about the general awareness of NBS among Czech mothers and identifies the variables associated with awareness. 

Information provision about NBS late in the pregnancy with a verbal reminder in the postnatal period shortly before sampling seems to be the most effective way of NBS education and retention of the information. Mothers should be informed from multiple sources, not only healthcare professionals but also new sources such as multimedia tools and electronic platforms, which have already proven to be effective [31]. Obstetricians and healthcare professional at neonatal departments should ascertain that primiparous and younger mothers without any previous experience of NBS should receive the appropriate information. 

Findings of this study have been used by the Czech Coordination Centre for NBS to enhance awareness of NBS amongst both the public and healthcare providers. This included nationwide educational seminars for healthcare professionals and the development and distribution of new educational materials in Czech and other language versions (English version shown in Suppl.). Future plans include using current communication media popular with the general public such as the development of an application for smartphones or the inclusion of information about NBS in existing applications about pregnancy. It is anticipated that using these modes of communication should ensure the target population is reached. The effectiveness of these different measures in enhancing NBS awareness amongst parents is something which could be evaluated at a future point using similar survey techniques or other appropriate methodologies. 

## Figures and Tables

**Figure 1 IJNS-05-00035-f001:**
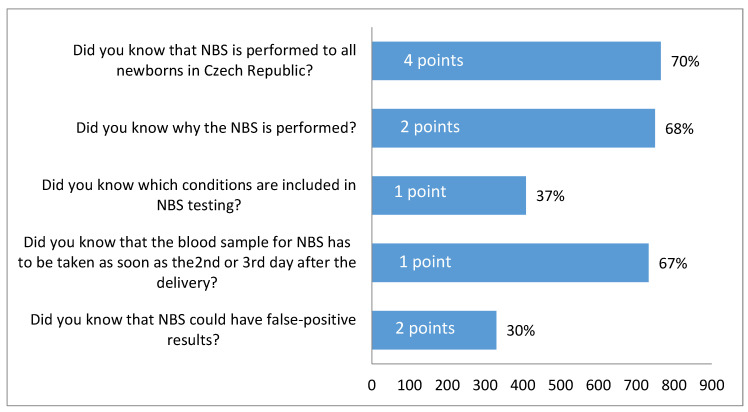
Participants’ responses to questions evaluating awareness about NBS. Corresponding points and percentage positive answers.

**Figure 2 IJNS-05-00035-f002:**
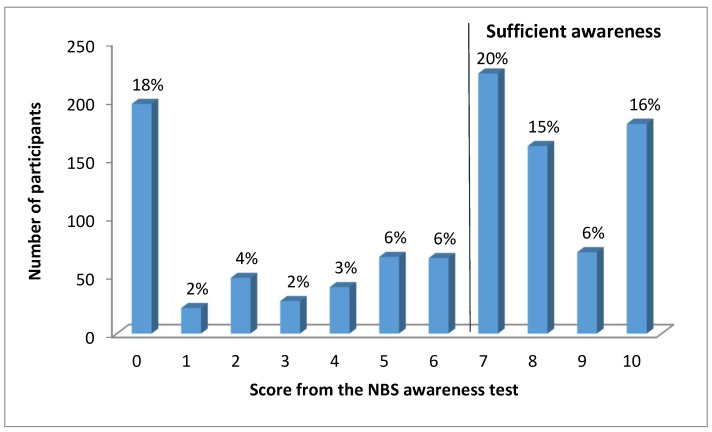
The distribution of scores from the questionnaire evaluating awareness about NBS.

**Figure 3 IJNS-05-00035-f003:**
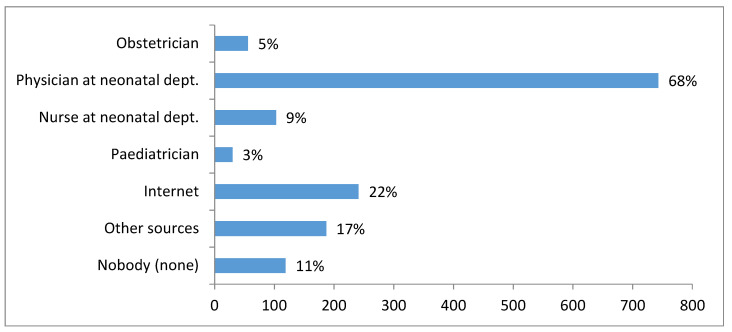
Sources of NBS information reported by participants.

**Figure 4 IJNS-05-00035-f004:**
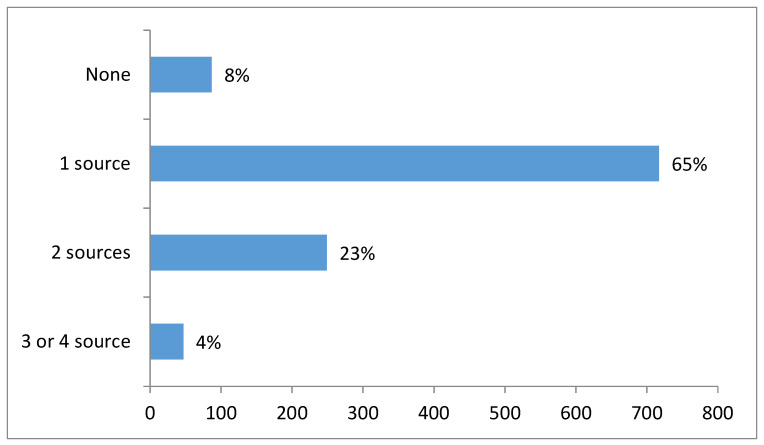
Number of sources from which participants received information about NBS.

**Table 1 IJNS-05-00035-t001:** Sample characteristics (*n* = 1100).

	*n*	This study	95 % CI	National data ^a^
**Age**				
≤29	345	31%	28.3–33.7	45%
≥30	755	69%	66.3–71.7	55%
**Education**				
Secondary school or less	126	12%	10.1–13.9	32%
High school graduate	442	40%	37.1–42.9	39%
University degree	532	48%	45.1–51.0	29%
**Parity**				
Primiparous	558	51%	48.1–53.9	47%
Multiparous	542	49%	46.1–51.9	53%
**Population of place of residence**				
<500	128	12%	10.1–13.9	
500–1000	67	6%	4.6–7.4	
1000–5000	160	14%	11.9–16.1	
5000–10,000	120	11%	9.2–12.8	
10,000–50,000	174	16%	13.8–18.2	
50,000–100,000	89	8%	6.4–9.6	
>100,000	110	10%	8.2–11.8	
>1 million (Prague)	252	23%	20.5–25.5	
**Current health status of participant**				
Feeling healthy	1002	91%	89.3–92.7	
With health complaints	98	9%	7.3–10.7	
**Complications during the pregnancy**				
No	869	79%	76.6–84.4	85%
Yes	231	21%	18.6–23.4	15%
**Child health status**				
Healthy child	1080	98%	97.2–98.8	98%
Child with a long-term serious illness	20	2%	1.2–2.8	2%
**Size of maternity hospital** (number of births per year)				
Small (<1000 births/year)	161	14.6%	12.5–16.7	16.9%
Medium (1000–3000 births/year)	463	42.1%	39.2–45.0	42.3%
Large (>3000 births/year)	473	43%	40.1–45.9	39.9%
Home	3	0.3%	2.0–4.0	0.09%

CI–confidence interval, ^a^ Report on Czech maternal and newborn population 2013 [16].

**Table 2 IJNS-05-00035-t002:** Analysis of the determinants associated with score from the questionnaire evaluating awareness about NBS (statistical significance *** *p* < 0.001; * *p* < 0.05).

	*n*	Score from Questionnaire Evaluating Awareness about NBS			*p*
		Mean	Median	Percentiles 25–75	
**Age**					0.001 ***
≤29	345	5.27	7.00	1.00–8.00	
≥30	755	6.02	7.00	4.00–8.00	
**Education**					N.S.
Secondary school or less	126	5.25	6.00	2.00–8.00	
High school graduate	442	5.69	7.00	2.00–8.00	
University degree	532	5.99	7.00	4.00–8.75	
**Parity**					0.001 ***
Primiparous	558	5.03	6.00	0.75–8.00	
Multiparous	542	6.56	7.00	5.00–9.00	
**Population of place of residence**					N.S.
<500	128	5.72	7.00	2.25–8.00	
500–1000	67	6.06	7.00	3.00–8.00	
1000–5000	160	5.70	7.00	2.25–8.00	
5000–10,000	120	5.18	7.00	1.25–8.00	
10,000–50,000	174	5.63	7.00	2.00–9.00	
50,000–100,000	89	5.90	7.00	2.50–10.00	
>100,000	110	5.86	7.00	3.00–8.25	
>1 million (Prague)	252	6.11	7.00	4.00–9.00	
**Current health status of participants (self-reported)**					N.S.
Feeling healthy	1002	5.80	7.00	3.00–8.00	
With health complaints	98	5.59	7.00	2.00–8.00	
**Complications during pregnancy**					N.S.
No	869	5.79	7.00	3.00–8.00	
Yes	231	5.76	7.00	3.00–8.00	
**Child health status**					0.028 *
Healthy child	1080	5.81	7.00	3.00–8.00	
Child with a long-term serious illness	20	4.10	4.50	0.00–7.00	
**Size of maternity hospital** (number of births per year)					0.041 *
Small (<1000 births/year)	161	5.76	7.00	3.00–8.00	
Medium (1000–3000 births/year)	463	6.04	7.00	3.00–9.00	
Large (>3000 births/year)	473	5.53	7.00	2.00–8.00	
**Number of sources of information**					0.001 ***
none	87	1.11	0.00	0.00–1.00	
One source	717	5.78	7.00	3.00–8.00	
≥2	296	7.17	8.00	7.00–9.00	
**Obstetrician as the source of information**					0.001 ***
Yes	56	7.66	8.00	7.00–9.00	
No	1044	5.68	7.00	2.00–8.00	

N.S.–Not significant.

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
