# Peer review of "Factors Influencing Parental Awareness about Newborn Screening"

_2409-515X, 2019, doi:10.3390/ijns5030035_

Round 1

Reviewer 1 Report

This manuscript surveys mothers in Czechoslovakia who bore children 3 months before in maternity units with regard to whether they were infomed about newborn screening (NBS), when they were given this information, who provided the information, and how much of the information did they recall. As in other similar surveys, the information was usually given in the immediate postnatal period when the mothers were inundated with other information and were less than optimally attentive recovering from the birth experience. They mothers generally were informed only the bare specifics of NBS, i.e., that the baby would get a heel stick and that it was to determine disorders that could be significant. Few of the mothers recalled receiving more specific information regarding the disorders being screened and so forth.

This survey information will be of much interest to those involved in NBS, including pediatricians or other physicians who care for newborns, the maternity units in which there are nurseries caring for neonates, those in public health who administer NBS, and many others involved either primarily or secondarily in NBS.

While the survey is not novel, it is unusual in its excellent plan, the comprehensiveness of the survey questions, and the superb analysis of the results.

Author Response

Dear Editor

Thank you very much for your responses and for the opportunity to submit a revised version of the manuscript. All comments from reviewer 2 have been carefully considered and the manuscript adjusted accordingly.

We are very grateful to the reviewers for their valuable feedback and input in order to produce an improved version of our manuscript. We have addressed the reviewers comments in the responses provided below.

Reviewers’ comments and suggestions:

Reviewer 1

Thank you for taking the time to review our paper. Your affirmative comments are appreciated. 

Reviewer 2 Report

This is a well executed study of NBS educational awareness in the Czech Republic. The findings are supported by other international studies, yet provide new information particularly within country. A few notes for consideration:

The authors conflate general NBS education and consent. They describe the unique issues with respect to consent in the Czech Republic, but there needs to be a bit more distinction between the processes, especially given that the survey appears to only address general education and not consent (figure 1 does not indicate that consent was part of the questionnaire). Please change the font color in the bars on figure 1 to white (or something lighter). What definition of false positive was provided to the participants? Were any cross-tab analyses performed? For example, were all the respondents who reported that no one informed them about NBS the same group who scored lowest in figure 2? In general, were there any correlations between scores and answers to the second survey?

Author Response

Dear Editor

Thank you very much for your responses and for the opportunity to submit a revised version of the manuscript. All comments from reviewer 2 have been carefully considered and the manuscript adjusted accordingly.

We are very grateful to the reviewers for their valuable feedback and input in order to produce an improved version of our manuscript. We have addressed the reviewers comments in the responses provided below.

Reviewers’ comments and suggestions:

Reviewer 2

Thank you for taking the time to review our paper and for your comments which will help to improve the quality of our manuscript.

Point 1: The authors conflate general NBS education and consent. They describe the unique issues with respect to consent in the Czech Republic, but there needs to be a bit more distinction between the processes, especially given that the survey appears to only address general education and not consent (figure 1 does not indicate that consent was part of the questionnaire)

Response:  Thank you for your feedback regarding the clarity between general education and consent. Indeed we did not specifically ask respondents in the questionnaire whether they consented to the procedure. In response to this we have reviewed the entire manuscript for the use of these terms and re-written the paragraph between lines 332-347 to make a clear distinction between the education and consent processes.

Point 2: Please change the font colour in the bars on figure 1 to white (or something lighter)

Response: The font colour has been changed. Thank you for this suggestion.

Point3: What definition of false positive was provided to the participants?

Response: The definition of a false positive as it appeared in the questionnaire has been added to the relevant paragraph. Thank you for this comment.

Point4: Were any cross-tab analyses performed? For example, were all the respondents who reported that no one informed them about NBS the same group who scored lowest in figure 2? In general, were there any correlations between scores and answers to the second survey?

Response: Thank you for raising these points about the analysis of the data.

The data in table 2 shows that all the respondents who reported that no one informed them about NBS also scored the lowest in the awareness questionnaire (Figure 2). A sentence has been added to Results section 3.4, line 224 to clarify this point. The information about the overlap between the group of mothers that were ≥30 years and multiparous has been added to the manuscript (lines 210-211).

Unfortunately the non-linear distribution of scores from the awareness test (Figure 2) did not allow us to perform regression analysis with demographic data from the second questionnaire used as explanatory variables. Thus after consultation with a statistician we performed either Mann-Whitney, Kruskal-Wallis or Χ2  analysis to evaluate the effect of individual demographic characteristics on the awareness score as shown in Table 2. We have also added this as a limitation of the study in the discussion (line 370).